# ADDRESSING COVARIATE SHIFTS WITH INFLUENCE AWARE ENERGY REGULARIZATION

## ABSTRACT

For classification problems where the classifier predicts $\bar{p}(y|\mathbf{x})$, namely the probability of label $y$ given data $\mathbf{x}$, an energy value can be defined (*e.g.* LogSumExp of the logits) and used to evaluate the estimated $\bar{p}(\mathbf{x})$ by the learned model, which is widely used for generative modeling and out-of-distribution (OOD) detection. In this paper, we identify a new promising direction that energy value on training data could be regularized for better generalization performance of the classifier facing covariate shift, as a principled means to address the shifts of $p(\mathbf{x})$, in various scenarios *e.g.*, long-tail recognition and domain generalization. Specifically, we propose to quantify the influence of regularizing energy value on the classification loss through the lens of influence function, a standard tool in robust statistics. This paves the way for our provably effective approach, Influence-Aware Energy Regularization (IAER), which aims at regularizing the energy value to adjust the decision margin and re-weight data samples. Experimental results demonstrate the efficacy of our method on several common benchmarks of class-imbalance classification and domain generalization. Source code will be made publicly available.

## 1 INTRODUCTION

General classification problems require the classifier to predict the conditional probability $\bar{p}(y|\mathbf{x})$ where $y$ is the label and $\mathbf{x}$ is the given data. Based on the $\bar{p}(y|\mathbf{x})$, an inherent energy value could be defined regarding the predicted $\bar{p}(\mathbf{x})$, known as energy models (LeCun et al., 2006). The energy model has been widely used for generative modeling (Grathwohl et al., 2020), out-of-distribution (OOD) detection (Liu et al., 2020), domain adaptation (Xie et al., 2022), *etc*. Specifically, in OOD scenarios, previous works (Liu et al., 2020; Xie et al., 2022) predominantly focus on the energy difference between in-distribution data (training data) and out-of-distribution (OOD) testing data such that in-distribution data would have lower energy than OOD data. However, the variation in energy value among in-distribution data points has been overlooked by previous works.

In this paper, we argue that the variation in energy values among training data could also influence the generalization performance of the classifier when facing covariate shift (Quinonero-Candela et al., 2008) where $p(y|\mathbf{x})$ stays the same while $p(\mathbf{x})$ changes.[1] The intuition behind is that the variation in energy values reflects the predicted $\bar{p}(\mathbf{x})$ which significantly affect the generalization performance when the true probability $p(\mathbf{x})$ shifts *e.g.* long-tail recognition (Wang et al., 2017), domain generalization (Blanchard et al., 2011), *etc*. With this high-level motivation, we aim at answering two follow-up research questions:

• Q1: How to quantify the influence of $\bar{p}(\mathbf{x})$, i.e., the energy value of data point, on the generalization performance on a shifted dataset (measured by loss on a held-out set)?

• Q2: How to leverage the recognized (implicit) influence among data to improve the model's generalization under covariate shifts?

To answer Q1, we resort to the influence function (Cook & Weisberg, 1982), a classic technique in robust statistics that can evaluate the influence of each training data point on the model's prediction. The seminal work Koh & Liang (2017) extends it to deep learning models and proposes to approximate

---

[1]Specifically, in this paper we mainly focus on imbalanced classification and domain generalization where the $p(\mathbf{x})$ shifts. Note that imbalanced classification is typically regarded as data selection bias while it could also be viewed as covariate shift in classification scenario where the $p(\mathbf{x})$ is reduced for $\mathbf{x}$ from the rare class.

the influence function with stochastic estimation. Instead of focusing on the weight of a certain data point, we propose to evaluate the influence of regularizing the energy on each data point upon the classifier's loss on the validation set, as an effective means to quantify the contribution to the generalization performance on a (proxy) set of new data.

Based on this, we devise a principled and generic method that fine-tunes the classifier with influence-aware energy regularization (Q2). We theoretically prove that when regularizing the energy value, it contributes to refining the model by re-weighting the samples in the training loss and adjusting the decision margin. We further empirically show that the estimated influence function can serve as an informative indicator that reflects the energy regularization associated with the generalization ability of the model. And pushing further, our proposed method can push the estimated $\bar{p}(\mathbf{x})$ closer to the real data distribution $p(\mathbf{x})$ when leveraging available prior knowledge of data distribution shifts in long-tailed recognition tasks. On top of these, experimental results demonstrate that our approach effectively improves the performance of classifiers on several class-imbalance learning and domain generalization benchmarks. **The highlights of this paper include:**

• To our best knowledge, this is the first work, both theoretically and empirically, to show that the variation of energy values among training data indicates the underlying estimated $\bar{p}(\mathbf{x})$ and could be regularized to influence the OOD generalization performance of the corresponding classifier.

• We extend the form of influence function to quantify the influence of regularizing the energy value on training data upon the classifier's loss on a (proxy) validation set (composed by training data).

• Using the extended influence function, we propose a principled and generic technique as a flexible plug-in to enhance the performance of the classifier using existing backbones. We theoretically prove that our method intrinsically re-weights the loss and adjusts the decision margin without affecting the optimal classifier. Its practical efficacy is verified on imbalanced classification and domain generalization against strong competitors.

• We further offer valuable insights regarding the usage of our quantified influence: 1) un-regularized energy values may contribute to the model's over-fitting; 2) the influence of energy regularization shows no correlation with the loss or the energy. The source code will be made publicly available.

## 2 PRELIMINARIES ON ENERGY FUNCTION AND INFLUENCE FUNCTION

In this section, we introduce notations and backgrounds related to this paper.

**The Classification Problem.** For a $K$-class classification problem, a parameterized classifier $f_\theta : \mathbb{R}^D \to \mathbb{R}^K$ maps data point $\mathbf{x} \in \mathbb{R}^D$ to $K$ real-valued logits where $\theta$ is the trainable parameter. For a data point $\mathbf{x}$ and its corresponding label $y$, the loss for the parameter $\theta$ is defined as $\mathcal{L}(\mathbf{x}, y, \theta)$.

**Energy-based Model for Data Distribution (LeCun et al., 2006; Grathwohl et al., 2020).** Energy based model $E(\mathbf{x}) : \mathbb{R}^D \to \mathbb{R}$ maps each data point $\mathbf{x}$ to a single, non-probabilistic scalar called the energy, where the energy value could be turned to a probability as:

$$p(\mathbf{x}) = \frac{e^{-E(\mathbf{x})/T}}{\int_{\mathbf{x}'} e^{-E(\mathbf{x}')/T}}. \tag{1}$$

For classifier $f_\theta$, the logits are typically converted to a normalized probability distribution with the Softmax function: $\bar{p}_\theta(y|\mathbf{x}) = \frac{\exp(f_\theta(\mathbf{x})[y])}{\sum_{y'} \exp(f_\theta(\mathbf{x})[y'])}$, where $f(\mathbf{x})[y]$ represents the logit corresponding to the $y$-th class. The joint distribution of data $\mathbf{x}$ and label $y$ could be defined as $\bar{p}_\theta(\mathbf{x}, y) = \frac{\exp(f_\theta(\mathbf{x})[y])}{Z(\theta)}$ where $Z(\theta)$ is unknown normalizing constant. By marginalizing out $y$, the unnormalized density model for $\mathbf{x}$ is $\bar{p}_\theta(\mathbf{x}) = \sum_y \bar{p}_\theta(\mathbf{x}, y) = \frac{\sum_y \exp(f_\theta(\mathbf{x})[y])}{Z(\theta)}$. Therefore the energy at data point $\mathbf{x}$ regarding to $\bar{p}_\theta(\mathbf{x})$ could be defined as:

$$E_\theta(\mathbf{x}) = -\log \sum_y \exp\left(f_\theta(\mathbf{x})[y]\right). \tag{2}$$

**Influence Function (see also definition in Cook & Weisberg (1982); Koh & Liang (2017)).** Given a training set with $n$ data points $D_{train} = \{(\mathbf{x}_1, y_1), (\mathbf{x}_2, y_2), \cdots (\mathbf{x}_n, y_n)\}$, the optimal parameter for empirical risk is given by $\hat{\theta} \stackrel{def}{=} \arg\min_\theta \frac{1}{n} \sum_{i=1}^n \mathcal{L}(\mathbf{x}_i, y_i, \theta)$. When the loss on a

training data point $(\mathbf{x}, y)$ is upweighted by a small $\epsilon$, the new optimal parameter becomes $\hat{\theta}_{\epsilon,(\mathbf{x},y)} = \arg\min_\theta (\frac{1}{n}\sum_{i=1}^n \mathcal{L}(\mathbf{x}_i, y_i, \theta) + \epsilon\mathcal{L}(\mathbf{x}, y, \theta))$. Assume that the empirical risk is twice-differentiable and strictly convex w.r.t. $\theta$, the influence function provides the influence of upweighting $(\mathbf{x}, y)$ on $\theta$:

$$\mathcal{I}_{\hat{\theta}}(\mathbf{x}, y) \overset{def}{=} \frac{\mathrm{d}\hat{\theta}_{\epsilon,(\mathbf{x},y)}}{\mathrm{d}\epsilon}|_{\epsilon=0} = -H_{\hat{\theta}}^{-1}\nabla_\theta\mathcal{L}(\mathbf{x}, y, \hat{\theta}). \tag{3}$$

where $H_{\hat{\theta}} = \frac{1}{n}\sum_{i=1}^n \nabla_\theta^2\mathcal{L}(\mathbf{x}_i, y_i, \hat{\theta})$ is the Hessian matrix. Using the chain rule, the influence of upweighting $(\mathbf{x}, y)$ on the loss at a valadation point $(\mathbf{x}_{va}, y_{va})$ (Koh & Liang, 2017) is:

$$
\begin{aligned}
\mathcal{I}_{(\mathbf{x}_{va}, y_{va})}(\mathbf{x}, y) &\overset{def}{=} \nabla_\theta\mathcal{L}(\mathbf{x}_{va}, y_{va}, \hat{\theta})^\top \frac{\mathrm{d}\hat{\theta}_{\epsilon,z}}{\mathrm{d}\epsilon}|_{\epsilon=0} \\
&= -\nabla_\theta\mathcal{L}(z_{test}, \hat{\theta})^\top H_{\hat{\theta}}^{-1}\nabla_\theta\mathcal{L}(\mathbf{x}, y, \hat{\theta}).
\end{aligned}
\tag{4}
$$

## 3 MAIN RESULTS AND APPROACH

In this section, we present our main approach and describe the proposed method. In Sec. 3.1, we prove that the energy value would arbitrarily shift during training without regularization. In Sec. 3.2, we define the influence function for energy regularization. In Sec. 3.3, we introduce our proposed method Influence Aware Energy Regularization(IAER). In Sec. 3.4, we analyze IAER and prove that our IAER would reweight data points and control the margin.

### 3.1 THE RELATIONSHIP BETWEEN THE ENERGY VALUE AND OUT OF DISTRIBUTION GENERALIZATION.

For a general case where the logits of the classifier are converted to a probability distribution with the softmax function, the logits could be viewed as a vector while the prediction of the classifier *i.e.* $\bar{p}(y|\mathbf{x})$ corresponds to the "direction" of the vector. The energy value as in Eq. 2 corresponds to the "norm" of the vector. Therefore the energy value does not directly affect the prediction of the classifier and is overlooked for classification problems.

**Proposition 3.1.** *[Arbitrary Energy] Consider* $\forall(\mathbf{x}, y) \in D_{train}$ *and a classifier* $f_\theta$. *For* $\forall\mathcal{E} \in \mathbb{R}$, *there exists a classifier* $g_\eta$ *that satisfy*

$$
\begin{aligned}
\bar{p}_\theta(y|\mathbf{x}) &= \bar{p}_\eta(y|\mathbf{x}), \\
E_\eta(\mathbf{x}) &= \mathcal{E}.
\end{aligned}
\tag{5}
$$

*where* $\bar{p}_\theta(y|\mathbf{x})$ *and* $\bar{p}_\eta(y|\mathbf{x})$ *is the conditional probability predicted by* $f_\theta$ *and* $g_\eta$ *respectively while* $E_\eta(\mathbf{x})$ *is the energy value of* $g_\eta$ *on data point* $\mathbf{x}$.

Proposition 3.1 shows that for a data point $\mathbf{x}$, the energy value $E_\theta(\mathbf{x})$ is not related to the prediction $\bar{p}_\theta(y|\mathbf{x})$. Even two classifiers with identical predictions could have different energy values.

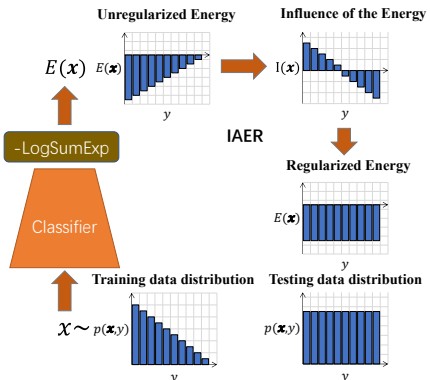

Figure 1: Illustration of our proposed Influence-Aware Energy Regularization (IAER) technique for long-tail classification. We quantify the influence of energy value regularization and apply regularization guided by the quantified influence. Our method could also be applied to other OOD scenarios where the data distribution shift is more implicit *e.g.* domain generalization.

However, we point out that the energy value could influence the generalization ability of the classifier under covariate shift where the data distribution $p(\mathbf{x})$ shifts. As illustrated in Fig. 1, by pushing the distribution $\bar{p}_\theta(\mathbf{x})$ corresponding to the energy $E_\theta(x)$ closer to the testing data distribution, the classifier could generalize better to the testing data distribution. While the distribution shift in imbalance classification setting (Buda et al., 2018; Cui et al., 2019) are explicit, the distribution shift in many scenarios *e.g.* domain adaptation, domain generalization are implicit. Therefore, we propose to quantify the influence of regularizing the energy value and further improve the generalization ability based on the influence. Detailed proofs are in the supplementary material (Appendix A).

## 3.2 INFLUENCE FUNCTION OF ENERGY REGULARIZATION

To evaluate the influence of energy regularization on the model performance on the testing set, we propose a metric using the influence function. We first study the change in model parameters brought by energy regularization. Specifically, consider adding a small positive regularization on the energy of a training data point $(\mathbf{x}_{tr}, y_{tr}) \in D_{train}$, the new optimized parameter becomes $\hat{\theta}_{\epsilon,(\mathbf{x}_{tr}, y_{tr})} = \arg\min_\theta (\frac{1}{n} \sum_{i=1}^n \mathcal{L}(\mathbf{x}_i, y, \theta) + \epsilon E_\theta(\mathbf{x}_{tr}))$. Similar to Eq. 3, the influence of regularizing energy $E_\theta(\mathbf{x}_{tr})$ on the parameter $\theta$ could be defined as:

$$\mathcal{I}_{\hat{\theta}}(\mathbf{x}_{tr}) \overset{def}{=} \frac{\mathrm{d}\hat{\theta}_{\epsilon,(\mathbf{x}_{tr}, y_{tr})}}{\mathrm{d}\epsilon}|_{\epsilon=0} = -H_{\hat{\theta}}^{-1} \nabla_\theta E_\theta(\mathbf{x}_{tr}). \tag{6}$$

where $H_{\hat{\theta}} = \frac{1}{n} \sum_{i=1}^n \nabla_\theta^2 \mathcal{L}(\mathbf{x}_i, y_i, \hat{\theta})$ is the Hessian matrix. Using the chain rule, the influence of regularizing energy $E_\theta(\mathbf{x}_{tr})$ on the loss at the validation point $\mathbf{x}_{val}$ is:

$$\begin{aligned}
\mathcal{I}_{\mathbf{x}_{val}}(\mathbf{x}_{tr}) &\overset{def}{=} \nabla_\theta \mathcal{L}(\mathbf{x}_{val}, \hat{\theta})^\top \frac{\mathrm{d}\hat{\theta}_{\epsilon,(\mathbf{x}_{tr}, y_{tr})}}{\mathrm{d}\epsilon}|_{\epsilon=0}, \\
&= -\nabla_\theta \mathcal{L}(\mathbf{x}_{val}, \hat{\theta})^\top H_{\hat{\theta}}^{-1} \nabla_\theta E_\theta(\mathbf{x}_{tr}).
\end{aligned} \tag{7}$$

## 3.3 MODEL FINETUNING WITH ENERGY REGULARIZATION

Based on our devised influence function of energy, we propose a principled method that introduces **I**nfluence **A**ware **E**nergy **R**egularization (we refer to it as **IAER**). It firstly calculates the average influence of energy regularization on a validation set $D_{val} = \{(\mathbf{x}_i^{val}, y_i^{val})\}_{i=1}^m$ for each training data point. To fairly compare with previous works, in our experiment, we take subsets of the training set as the validation set without introducing any additional data.

$$\mathcal{I}_{val}(\mathbf{x}_i^{tr}, y_i^{tr}) = \frac{1}{m} \sum_{j=1}^m \mathcal{I}_{(\mathbf{x}_j^{val}, y_j^{val})}(\mathbf{x}_i^{tr}, y_i^{tr}). \tag{8}$$

To reduce the loss on the validation set, we should increase the energy on the data points with a positive influence of energy regularization and decrease the energy on those with a negative influence. Therefore, we finetune the model with energy penalties determined by the corresponding influence value. The loss with energy regularization on a training batch $\mathcal{B} = \{z_i = (x_i, y_i)\}$ is:

$$\mathcal{L}'(\mathbf{x}_i^{tr}, y_i^{tr}, \theta) = \frac{1}{|\mathcal{B}|} \sum_{\mathbf{z_i} \in \mathcal{B}} \left( \mathcal{L}_{ce}(\mathbf{x}_i^{tr}, y_i^{tr}, \theta) + \hat{\beta}_\mathbf{x} \cdot E_\theta(\mathbf{x}_i^{tr}) \right), \tag{9}$$

$$\text{where } \hat{\beta}_\mathbf{x} = -\beta \cdot \mathcal{I}_{val}(\mathbf{x}_i^{tr}, y_i^{tr})/\mathcal{I}_{val}^{max}.$$

Here $\mathcal{I}_{val}^{max}$ is the maximum absolute value of influence value over the train set $D_{train}$ and we set the hyperparameter $0 < \beta < 1$, by which we make sure that $\hat{\beta}_\mathbf{x} < 1$. For the detailed algorithm, please refer to Alogrithm 1 in Appendix B.

## 3.4 ENERGY REGULARIZATION AS REWEIGHTING AND MARGIN CONTROL

For a data point $(\mathbf{x}, y)$, assume the loss for classifier $f_\theta$ is:

$$\mathcal{L}(\mathbf{x}, y, \theta) = \mathcal{L}_{ce}(\mathbf{x}, y, \theta) + \hat{\beta}_\mathbf{x} \cdot E_\theta(\mathbf{x}). \tag{10}$$

where $\mathcal{L}_{ce}(\mathbf{x}, y, \theta) = -\log \frac{\exp(f_\theta(\mathbf{x})[y])}{\sum_{y'} \exp(f_\theta(\mathbf{x})[y'])}$ is the cross-entropy loss and $\hat{\beta}_\mathbf{x} \in \mathcal{R}$ is the coefficient for the energy regularization. Then the gradient of the loss is:

$$\frac{\partial \mathcal{L}(\mathbf{x}, y, \theta)}{\partial \theta} = \left[ (1 - \hat{\beta}_\mathbf{x}) \cdot \bar{p}(y|\mathbf{x}) - 1 \right] \frac{\partial f_\theta(\mathbf{x})[y]}{\partial \theta} + (1 - \hat{\beta}_\mathbf{x}) \sum_{y' \neq y} \bar{p}(y'|\mathbf{x}) \cdot \frac{\partial f_\theta(\mathbf{x})[y']}{\partial \theta}. \tag{11}$$

When $\hat{\beta}_\mathbf{x} \neq 1$, we could further derive the gradient as:

$$\frac{\partial \mathcal{L}(\mathbf{x}, y, \theta)}{\partial \theta} = (1 - \hat{\beta}_\mathbf{x}) \cdot \left[ \bar{p}(y|\mathbf{x}) - \frac{1}{1 - \hat{\beta}_\mathbf{x}} \right] \frac{\partial f_\theta(\mathbf{x})[y]}{\partial \theta} + (1 - \hat{\beta}_\mathbf{x}) \cdot \sum_{y' \neq y} \bar{p}(y'|\mathbf{x}) \cdot \frac{\partial f_\theta(\mathbf{x})[y']}{\partial \theta}. \tag{12}$$

As shown in Eq. 12, the influence of energy regularization is two folds: adjusting the margin[2] and reweighting data points. When $0 < \hat{\beta}_{\mathbf{x}} < 1$, the energy regularization enlarges the margin by pushing down the coefficient of $\frac{\partial f_\theta(\mathbf{x})[y]}{\partial \theta}$ and down-weights the data point $\mathbf{x}$ with coefficient $(1 - \hat{\beta}_{\mathbf{x}})$. When $\hat{\beta}_{\mathbf{x}} < 0$, the regularizer reduces the margin and up-weights the data $\mathbf{x}$ with coefficient $(1 - \hat{\beta}_{\mathbf{x}})$. Beyond that, we further highlight that adding regularization would not interfere the optimal classifier.

**Lemma 3.2.** *[Unaffected Optimal] If there exist an optimal model $f_\theta^*$ that minimizes the negative log likelihood loss. Then for any model $g_\upsilon$, there always exits a model $h_\xi$ that satisfies:*

$$\forall \mathbf{x} \in \mathcal{R}^D, h_\xi(\mathbf{x}) = f_\theta^*(\mathbf{x}), E_\upsilon(\mathbf{x}) = E_\xi(\mathbf{x}). \tag{13}$$

It indicates that if there is an optimal classifier $f_\theta^*$ for the loss with energy regularization $\mathcal{L}(\mathbf{x}, y, \theta)$, it is the optimal classifier for negative log likelihood loss $\mathcal{L}_{ce}(\mathbf{x}, y, \theta)$. The proof is in Appendix A.

## 4 EXPERIMENT

In this section, we conduct experiments on long-tail recognition benchmarks and domain generalization benchmarks. In Sec. 4.1, we validate that energy regularization could influence the testing performance of the classifier and the influence function could reflect the influence. In Sec. 4.2, we provide empirical evidence showing that our proposed method IAER would push the estimated data distribution closer to the actual data distribution on long-tail datasets. In Sec. 4.3 and Sec. 4.4, we verify the effectiveness of our proposed method IAER on long-tail recognition benchmarks and domain generalization benchmarks. Note that when applying the IAER, we estimate the influence solely based on training data to fairly compare with baselines. In Sec. 4.5, we investigate the relationship between the influence of the energy regularization and other factors (*e.g.* the training loss).

### 4.1 VALIDATION OF ESTIMATED INFLUENCE OF ENERGY REGULARIZATION

To quantify the influence, we approximate the influence function of the energy value using stochastic approximation following Koh & Liang (2017) over the whole testing set. Then we finetune the pretrained model with energy regularization on one of the data points and evaluate the change in testing loss compared to finetuning the model without any energy penalties. Because batch normalization may affect the energy, we fix the parameter for batch normalization during the finetune.

Specifically, we take a small CNN architecture with a depth of 6 (Basu et al., 2021) trained on long-tailed CIFAR10 (Cui et al., 2019) with ERM and LDAM (Cao et al., 2019) as the pretrained model and the model is finetuned for 50 epochs. We follow Koh & Liang (2017) to validate the approximation of influence function of energy regularization. As shown in Fig. 5, the influence function is highly correlated to the actual change in testing loss (Pearson's R $= 0.9154$ for CNN pretrained with ERM and Pearson's R $= 0.9037$ for CNN pretrained with LDAM). It shows that energy regularization could influence the testing loss of the classifier and that the influence function could reflect the influence. For more details and results, see Appendix C

### 4.2 RELATION BETWEEN ESTIMATED DISTRIBUTION AND CLASSIFIER PERFORMANCE

**Pushing the estimated $\bar{p}(\mathbf{x})$ closer to the actual testing distribution $p(\mathbf{x})$ could improve the performance of the classifier.** We plot the average influence of energy regularization of each class on the small CNN trained on CIFAR10-LT. As shown in Fig. 2, the average influence of energy regularization of different class is positively correated to the number of data points of the corresponding class (The Pearson's R is $0.70$ for long-tailed CIFAR10 with imbalance ratio at $10$). The lower the influence of the energy regularization means that the testing loss of the classifier would be lower after adding a positive energy regularization. It indicates that pushing down the energy value of data points of less frequent class and pulling up the energy value of data points of more frequent class would boost the testing performance *i.e.* pull up the predicted $\bar{p}(\mathbf{x})$ for less frequent class and push down the predicted $\bar{p}(\mathbf{x})$ for more frequent class. Since the probability density $p(\mathbf{x})$ of data points of less frequent class is much lower and the $p(\mathbf{x})$ of data points of more frequent class is much higher in the imbalanced training set compared to the testing set, it shows that pushing the predicted $\bar{p}(\mathbf{x})$ closer to the real $p(\mathbf{x})$ of the testing set would boost the testing performance. According to

---

[2]Following Cao et al. (2019), the margin here means $f(x)_y - max_{j \neq y} f(x)_j$ where $y$ is the label.

Table 1: Average testing error (%) of two variants of our method on Imbalanced CIFAR10/CIFAR100.

| Dataset | Imbalanced CIFAR10 | | | | Imbalanced CIFAR100 | | | |
|---|---|---|---|---|---|---|---|---|
| Imbalance type | long-tailed (Cui et al., 2019) | | step (Buda et al., 2018) | | long-tailed (Cui et al., 2019) | | step (Buda et al., 2018) | |
| Imbalance Ratio | 100 | 10 | 100 | 10 | 100 | 10 | 100 | 10 |
| ERM (Cao et al., 2019) | 29.64 | 13.39 | 36.70 | 15.73 | 61.68 | 43.41 | 61.45 | 45.30 |
| LDAM-DRW (Cao et al., 2019) | 22.84 | 12.38 | 24.64 | 12.58 | 57.96 | 43.33 | **54.64** | 43.16 |
| ERM + IAER | 24.17 | 12.98 | 28.67 | 14.38 | 60.40 | **42.41** | 60.88 | 44.90 |
| LDAM-DRW + IAER | **21.63** | **12.28** | **24.11** | **12.30** | **57.19** | 43.33 | 55.39 | **43.10** |

Sec. 3.4, our regularization method, generally, enlarges the margin and down-weights for the less frequent classes. On the other hand, it reduces the margin and up-weights for the more frequent classes. For more results, refer to Appendix C.

### 4.3 FINETUNING THE CLASSIFIER ON IMBALANCE DATA

We evaluate IAER on the imbalanced version of CIFAR10, CIFAR100 (Cui et al., 2019) and ImageNet-LT (Liu et al., 2019) that are artificially created with class imbalance and iNaturalist 2018 (Van Horn et al., 2018), a naturally long-tailed dataset. Experiments are conducted on imbalanced CIFAR following Cao et al. (2019) and on ImageNet-LT following Kang et al. (2020). For fair comparison, the validation set used to calculate influence function is sampled from the training set and the models are not exposed to testing data during training.

#### 4.3.1 RESULTS ON CIFAR

**Baselines.** We evaluate IAER (Alg. 1) with ResNet-32 pretrained by: 1) Empirical risk minimization (ERM): with equal weight for each training data, and the model is trained to minimize the cross-entropy. 2) LDAM-DRW (Cao et al., 2019): LDAM introduces a label-distribution aware margin loss which enlarges the decision while DRW applies re-weighting or re-sampling after the last learning rate decay. Since ERM is the basic training algorithm and a typical baseline while LDAM-DRW achieves SOTA on imbalanced CIFAR datasets, we take these two methods as baselines.

**Datasets.** CIFAR10 and CIFAR100 both contain $50,000$ images in training and $10,000$ images in testing with 10 and 100 classes, respectively. We construct the imbalanced version of CIFAR10 and CIFAR100 by reducing the number of images for each class. Two types of imbalance are considered: long-tailed imbalance (Cui et al., 2019) and step imbalance (Buda et al., 2018). For long-tailed imbalance, the number of data points follows an exponential decay across different classes. For step imbalance, data points in half of the classes are reduced to the same number while the number of data points in the other classes remains the same. The imbalance ratio of the imbalanced dataset is defined as the ratio between the maximum number of data points in one class and the minimum number of data points in one class.

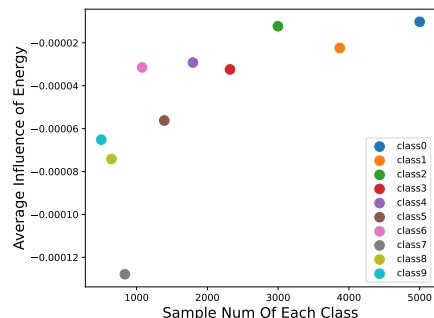

Figure 2: We average the influence of energy regularization on the CNN trained on CIFAR10-LT over the training data points of each class, and plot it against the number of data points for each class in long-tail CIFAR10 with imbalance ratio at 10.

**Implementation.** Note that IAER requires calculating the influence function with a validation set. To fairly compare with the baselines, we randomly sample a class-balanced subset from the training set to compose the validation set where the number of data points per class is determined by the minimum number of data points per class in the training set. We train each model for 200 epochs and finetune it for 5 epochs. The error rate at the last epoch is reported. Refer to Appendix B for details.

**Results.** As shown in Table 1, our method could effectively boost the testing performance after only 5 epochs. The more imbalanced the more effective is IAER and IAER could greatly improve the ERM pretrained model. For instance, IAER reduces the testing error of ERM pretrained model for $5.47\%$ (from $29.64\%$ to $24.17\%$) on long-tailed CIFAR10 with the imbalance ratio at 100. For LDAM-DRW pretrained model, IAER can also improve the testing performance on CIFAR10.

For CIFAR100, IAER also improves the testing performance for ERM pretrained model and improves the LDAM-DRW pretrained model on long-tailed CIFAR100 with the imbalance ratio at 100 and step

Table 2: Experiments on ImageNet-LT and iNaturalist. The validation set for IAER is composed by images of different class in train set.

| Dataset and model Method | ResNeXt-50 on ImageNet-LT | | | | ResNet-152 on iNatrualist | | | |
|---|---|---|---|---|---|---|---|---|
| | Many | Median | Few | All | Many | Median | Few | All |
| cRT (Kang et al., 2020) | 61.8 | 46.2 | 27.4 | 49.6 | 75.9 | 71.9 | 69.1 | 71.2 |
| cRT + IAER[Few] | 61.0 | 45.6 | **29.1** | 49.3 | 76.1 | 71.6 | **69.5** | 71.2 |
| cRT + IAER[Median] | 58.5 | **48.7** | 26.0 | 49.4 | 75.8 | **72.3** | 68.0 | 71.0 |
| cRT + IAER[Many] | **62.7** | 44.5 | 26.9 | 49.1 | **77.8** | 69.9 | 66.5 | 69.4 |
| LWS (Kang et al., 2020) | 60.2 | 47.2 | 30.3 | 49.9 | 74.3 | 72.4 | 71.2 | 72.1 |
| LWS + IAER[Few] | 60.1 | 47.1 | **32.1** | **50.1** | 74.4 | 72.4 | **71.6** | **72.3** |
| LWS + IAER[Median] | 58.1 | **49.0** | 30.3 | 50.0 | 74.5 | **72.8** | 70.9 | 72.2 |
| LWS + IAER[Many] | **61.5** | 45.5 | 30.1 | 49.6 | **74.9** | 72.6 | 71.0 | 72.2 |

imbalanced CIFAR100 with the imbalance ratio at 10. However, the improvement on imbalanced CIFAR100 brought by our IAER is much smaller than that on imbalanced CIFAR10. We conjecture that the calculated influence of energy regularization on our sampled validation set for CIFAR100 is less accurate since the number of images per class in CIFAR100 is much smaller than that of CIFAR10 *e.g.* only 5 images for the least frequent class when imbalance ratio is 100.

### 4.3.2 RESULTS ON IMAGENET-LT AND INATRUALIST 2018

**Baselines.** We evaluate IAER with ResNeXt-50 (Xie et al., 2017) pretrained by the techniques and protocols proposed in Kang et al. (2020) where each model is divided into two parts: backbone and linear classifier. The protocol include (1) Classifier Re-training (cRT): employ the backbone trained with ERM and retrain the linear classifier with the class balance sampling method. (2) Learnable Weight Scaling (LWS): Rescale the weight of the classifier for each class by a rescale factor learned with the class balance sampling method as in cRT.

**Datasets.** Experiments for large scale long-tail datasets are conducted on ImageNet-LT (Liu et al., 2019) and iNaturalist 2018 (Van Horn et al., 2018). ImageNet-LT is artificially truncated from ImageNet (Deng et al., 2009), where the label distribution follows a long-tailed distribution. It has 1000 classes and the number of images per class ranges from 1280 to 5 images. iNatualist 2018 is a real-world, long-tailed dataset with 8142 classes. We follow Liu et al. (2019) and report the testing accuracy on three kinds of class sets: *Many-shot* (over 100 images), *Medium-shot* ($20 \sim 100$ images) and *Few-shot* (less than 20 images). The testing accuracy on all classes is denoted as *All*.

**Implementation.** Note that the minimum number of data points per class in the training set of iNaturalist is 2, which makes the possible class-balanced subset of the training set extremely small. Therefore, for ImageNet-LT and iNaturalist, we take images of *Many-shot*, *Medium-shot* and *Few-shot* classes in the training set as the validation set respectively. We employ the backbone provided by Kang et al. (2020) and finetune or retrain the classifier. For ImageNet-LT, we calculate the influence of energy regularization on the ResNeXt-50 pretrained for 90 epochs and retrain the classifiers with energy regularization as proposed in Algorithm 1 for 10 epochs. For iNaturalist 2018, we calculate the influence of energy regularization of the ResNet-152 pretrained for 200 epochs. The classifiers on the iNaturalist are retrained for 30 epochs with energy regularization. More details are in Appendix B.

**Results.** As shown in Table 2, IAER[Few] means that the validation set used for calculating influence is composed by images of *few-shot* classes in the training set while IAER[Median] and IAER[Many] means the validation set is composed by *median-shot* classes and *many-shot* classes respectively. For ImageNet-LT and iNatrualist 2018, we could observe that the accuracy on the classes used to calculate the influence function is boosted *e.g.* IAER[Few] boosts the accuracy of the classifier on *few-shot* classes while IAER[Median] boosts the accuracy of the classifier on *median-shot* classes. LWS combined with IAER[Few] could improve the accuracy on the whole testing set. However, due to the limitation of the validation set the improvement is small.

### 4.4 RESULTS OF DOMAIN GENERALIZATION

**Baselines.** We follow DomainBed (Gulrajani & Lopez-Paz, 2021) to conduct experiments for domain generalization. DomainBed is a testbed for domain generalization. As Gulrajani & Lopez-Paz (2021) shows that ERM outperforms SOTAs by average performance on common benchmarks as evaluated

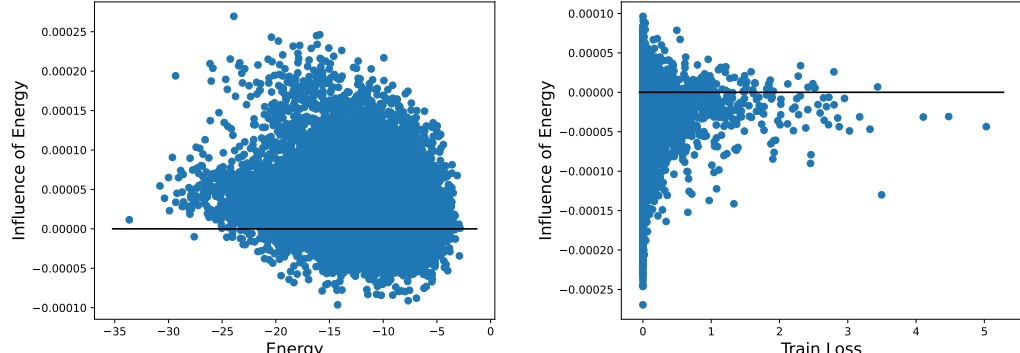

(a) Influence of energy regularization against the energy  (b) Influence of energy regularization against the loss

Figure 3: Influence of energy regularization, loss, and energy at each training data point (a dot in plot) in long-tailed CIFAR10 with the imbalanced ratio at 10 for ResNet-32 trained with ERM. We plot the influence of energy regularization against the loss and energy.

Table 3: Experiment results for domain generalization conducted based on DomainBed (Gulrajani & Lopez-Paz, 2021) on CMNIST (Arjovsky et al., 2019), PACS (Li et al., 2017) and VLCS (Fang et al., 2013).(* means we use the results from (Gulrajani & Lopez-Paz, 2021))

| Method | CMNIST | PACS | VLCS |
|---|---|---|---|
| ERM (Vapnik, 1998) | $51.5 \pm 0.1$ | $85.5 \pm 0.1$ | $77.4 \pm 0.2$ |
| IRM (Arjovsky et al., 2019)* | $52.0 \pm 0.1$ | $83.5 \pm 0.8$ | $78.5 \pm 0.5$ |
| GroupDRO (Sagawa et al., 2019)* | $52.0 \pm 0.0$ | $84.4 \pm 0.8$ | $76.7 \pm 0.6$ |
| MLDG (Li et al., 2018a)* | $51.5 \pm 0.1$ | $84.9 \pm 1.0$ | $77.2 \pm 0.4$ |
| CORAL (Sun & Saenko, 2016) | $51.2 \pm 0.1$ | $86.1 \pm 0.2$ | $78.8 \pm 0.6$ |
| SagNet (Nam et al., 2021) | $51.7 \pm 0.0$ | $86.3 \pm 0.1$ | $77.8 \pm 0.5$ |
| ERM + IAER | $51.9 \pm 0.1$ | $86.6 \pm 0.1$ | $78.5 \pm 0.2$ |

in a consistent and realistic setting, we combine our method with ERM, and take the algorithms implemented in DomainBed as baselines.

**Datasets.** Experiments are performed on benchmarks ColoredMNIST (Arjovsky et al., 2019), PACS (Li et al., 2017) and VLCS (Fang et al., 2013). We follow the setting in DomainBed (Gulrajani & Lopez-Paz, 2021) to process the data.

**Implementation.** As mentioned in Sec. 4.3, we take the training set as the validation set to calculate the influence function for a fair comparison. We follow the setting in DomainBed (Gulrajani & Lopez-Paz, 2021) to train the model. We demonstrate the results using training domain validation as model selection criteria which use a validation set sampled from the training domain for model selection. For each algorithm and testing domain, we conduct a random search of 5 trails. For more details, see Appendix B.

**Results.** As shown in Table 3, our IAER could improve the accuracy on the test domain without the test domain information. It implies that regularizing energy on the training domains helps the generalization from the training domains to the testing domain. More results are in Appendix C.

### 4.5 INFLUENCE OF ENERGY REGULARIZATION

In this section, we take a closer look on the influence of energy regularization and provide some insights. We conduct experiment on the ResNet-32 trained on the imbalanced CIFAR10. For more details and more results on imbalanced CIFAR100, please refer to Appendix B.

**The influence of energy regularization is not correlated to the energy value and the training loss.** As shown in Fig. 3(a), we find that generally the influence of energy regularization on the training data point is not related to the energy value of the data point (Pearson's R is $-0.21$). As for the training loss, Fig. 3(b) shows that the influence of energy regularization on the data points of similar training loss ranges from positive to negative. The influence of energy regularization is not related to the training loss (Pearson's R is $-0.04$). *It indicates that one could not predict the influence of energy regularization on the data point based on the training loss or the energy value.*

**Energy regularization have large influence mainly on well-classified data points.** As shown in Fig. 3, the range of the influence of energy regularization expands as the training loss decreases. Therefore, data points where energy regularization have a large influence are generally well-classified data points with low loss. As pointed out in Remark A.1, the energy value is unstable during the training, *we conjecture that the un-regularized energy value of well-classified data points is one of the possible reasons for the overfitting of the classifier*.

## 5 RELATED WORKS AND FURTHER DISCUSSION

**Energy Based Learning.** Energy-based models (EBMs) (LeCun et al., 2006; Ranzato et al., 2006; 2007) provide a unified theoretical framework for various learning models. EBMs associate a scalar energy value to each configuration of the variables where the energy value is lower for observed configurations than the unobserved ones. Xie et al. (2016) shows that a generative random field model can be derived from a discriminative neural network. While Grathwohl et al. (2020) finds that neural classifiers are also energy-based models for joint distribution and devise a hybrid model that acts as both discriminative and generative models. Liu et al. (2020) proposes to use the energy value to detect OOD samples, which has been theoretically proved (Bitterwolf et al., 2022) to be equal to training an additional binary discriminator. The influence of this specific energy on the discriminative model itself is often simply assumed as the lower energy value on the training set the better (Zhao et al., 2022). Recent work (Xie et al., 2022) minimizes the distance of energy distribution between the source domain and target domain to enhance the performance in domain adaptation.

**Long-Tail Recognition.** Long-tail recognition has drawn increasing attention (Wang et al., 2017; Zhou et al., 2018; Liu et al., 2019; Zhong et al., 2019; He et al., 2021; Zhong et al., 2021) due to the pervasiveness of the imbalanced data in real-world scenarios. Most methods could be divided into three categories: re-sampling the data, re-weighting the loss, and transfer learning. For re-sampling, various methods have been proposed to re-sample the dataset to achieve a more balanced data distribution (Chawla et al., 2002; Estabrooks et al., 2004; Han et al., 2005; Liu et al., 2009; Shen et al., 2016; Wang et al., 2020; Zhang & Pfister, 2021). As for re-weighting, re-weighting methods assign different losses to different classes (Zhang et al., 2018; Zhao et al., 2019; Ye et al., 2020; Hsieh et al., 2021) or different data samples (Lin et al., 2017; Ren et al., 2018; Shu et al., 2019) to achieve a more balanced performance on each class. Specifically, LDAM (Cao et al., 2019) proposes a distribution aware loss that enlarges the margin to less frequent (tail) classes.

**Domain Generalization.** Specifically, domain generalization (DG) (Blanchard et al., 2011) aims to train a model using data from a single or multiple source domains that would generalize well to any out-of-distribution(OOD) target domains. Various methods have been proposed to tackle the domain generalization problem including learning domain-invariant representations (Muandet et al., 2013; Li et al., 2018b;c), augmenting the data (Zhou et al., 2020; Yan et al., 2020) and applying meta-learning to domain generalization (Li et al., 2018a; Balaji et al., 2018). See Wang et al. (2021); Zhou et al. (2021) for a comprehensive survey.

**Discussion of the limitation.** One may argue that the influence function is derived under the assumption that the loss is convex, which is violated in deep learning scenario, and the approximation of the influence function could be inaccurate (Stephenson et al., 2020; Basu et al., 2021). In this paper, we validate that the approximated influence function of energy regularization is highly positively correlated to the actual change in the testing loss for a small CNN (Basu et al., 2021) on CIFAR10 and CIFAR100 in Sec. 4.1. For large datasets (*e.g.* ImageNet-LT and iNatrualist2018) and large models, our results in Sec. 4.3 and Sec. 4.4 shows the effectiveness of our method. Still, imporvement could be brought by a more stable tool to quantify the influence of energy regularization.

## 6 CONCLUSION

We have proposed to study the influence of energy regularization on the generalization ability of the model. Based on the influence function, which is a classic method in robust statistics, we manage to quantify the influence and propose an influence-aware energy regularizer for the energy value on training data point according to the quantified influence. Theoretically, we prove that our method could re-weight the loss and affect the margin without affecting the optimal model. Empirically, we show that the proposed quantification correctly reflects the influence of energy regularization and we verify the effectiveness on various imbalanced datasets and domain generalization benchmarks.

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

# A PROOFS

## A.1 PROOF FOR PROPOSITION 3.1

*Proof.* For the classifier $f_\theta : \mathbb{R}^D \to \mathbb{R}^K$, assume the energy on data point $\mathbf{x} \in \mathcal{R}^D$ is $E_\theta(\mathbf{x})$. For $\forall \mathcal{E} \in \mathcal{R}$, a classifier $g_\eta : \mathcal{R}^D \to \mathcal{R}^K$ could be defined that satisfies:

$$\forall i \in \{1, 2, \cdots, K\}, \quad g_\eta(\mathbf{x})[i] = f_\theta(\mathbf{x})[i] - \mathcal{E} + E_\theta(\mathbf{x}). \tag{14}$$

By adding a certain value $E_\theta(\mathbf{x}) - \mathcal{E}$ to each logit $f_\theta(\mathbf{x})[i]$, the prediction of $g_\eta$ is the same as the prediction of $f_\theta$ while the energy value of $g_\eta$ is changed to $\mathcal{E}$.

For the predicted $\bar{p}_\eta(y|\mathbf{x})$ we have:

$$\begin{aligned}
\bar{p}_\eta(y|\mathbf{x}) &= \frac{\exp\left[f_\theta(\mathbf{x})[y]\right] - \mathcal{E} + E_\theta(\mathbf{x})]}{\sum_i \exp\left[f_\theta(\mathbf{x})[i]\right) - \mathcal{E} + E_\theta(\mathbf{x})]}, \\
&= \frac{\exp\left[f_\theta(\mathbf{x})[y]\right)]}{\sum_i \exp\left[f_\theta(\mathbf{x})[i]\right)]}, \\
&= \bar{p}_\theta(y|\mathbf{x}).
\end{aligned} \tag{15}$$

For the energy $E_\eta(\mathbf{x})$ on the $g_\eta$, we have:

$$\begin{aligned}
E_\eta(\mathbf{x}) &= -\log \sum_i \exp\left[g_\eta(\mathbf{x})[i]\right] \\
&= -\log \sum_i \exp\left[f_\theta(\mathbf{x})[i] - \mathcal{E} + E_\theta(\mathbf{x})\right] \\
&= -\log\left(\exp[E_\theta(\mathbf{x}) - \mathcal{E}] \cdot \sum_i \exp\left[f_\theta(\mathbf{x})[i]\right]\right) \\
&= \mathcal{E} - E_\theta(\mathbf{x}) - -\log \sum_i \exp\left[f_\theta(\mathbf{x})[i]\right] \\
&= \mathcal{E} - E_\theta(\mathbf{x}) + E_\theta(\mathbf{x}) \\
&= \mathcal{E}.
\end{aligned} \tag{16}$$

$\square$

## A.2 UNSTABLE ENERGY VALUE DURING TRAINING

*Remark* A.1. [**Uncertain Energy Shift**] When optimizing the classifier $f_\theta$ with negative log-likelihood loss $\mathcal{L}_{ce}(\mathbf{x}, y, \theta)$, the gradient of the parameter $\theta$ is:

$$\frac{\partial \mathcal{L}_{ce}(\mathbf{x}, y, \theta)}{\partial \theta} = -\frac{\partial E_\theta(\mathbf{x})}{\partial \theta} - \frac{\partial f_\theta(\mathbf{x})[y]}{\partial \theta}. \tag{17}$$

where $-\frac{\partial E_\theta(\mathbf{x})}{\partial \theta}$ pulls up the energy and $-\frac{\partial f_\theta(\mathbf{x})[y]}{\partial \theta}$ pushs down the energy. Therefore the energy $E_\theta(\mathbf{x})$ may fluctuate regarding the output of the model as the loss decreases.

*Proof.* The negative log likelihodd loss is:

$$\begin{aligned}
\mathcal{L}_{nll}(\mathbf{x}, y, \theta) &= -\log \frac{\exp(f_\theta(\mathbf{x})[y])}{\sum_i \exp(f_\theta(\mathbf{x})[i])} \\
&= \log \sum_i \exp(f_\theta(\mathbf{x})[i]) - f_\theta(\mathbf{x})[y]) \\
&= -E_\theta(\mathbf{x}) - f_\theta(\mathbf{x})[y]).
\end{aligned} \tag{18}$$

Then gradient of negative log likelihood loss *w.r.t* parameter $\theta$ is:

$$\begin{aligned}
\frac{\partial \mathcal{L}_{nll}(\mathbf{x}, y, \theta)}{\partial \theta} &= \frac{\partial[-E_\theta(\mathbf{x}) - f_\theta(\mathbf{x})[y])]}{\partial \theta} \\
&= -\frac{\partial E_\theta(\mathbf{x})}{\partial \theta} - \frac{\partial f_\theta(\mathbf{x})[y])}{\partial \theta}.
\end{aligned} \tag{19}$$

---

**Algorithm 1** Finetune the classifier with energy regularization.

---

**Input:** Training Set $D_{train} = \{(\mathbf{x}_i^{tr}, y_i^{tr})\}_{i=1}^n$. Validation Set $D_{val} = \{(\mathbf{x}_i^{val}, y_i^{val})\}_{i=1}^m$. Pre-trained model $f_\theta$, epoch $T$, batch size $r$, learning rate $\alpha$, scale factor $\beta$
**for** $(\mathbf{x}_i^{tr}, y_i^{tr})$ **in** $D_{train}$ **do**
    $\mathcal{I}_{val}(\mathbf{x}_i^{tr}, y_i^{tr}) \leftarrow \frac{1}{m} \sum_{j=1}^m \mathcal{I}_{(\mathbf{x}_j^{val}, y_j^{val})}(\mathbf{x}_i^{tr}, y_i^{tr})$
**end for**
$\mathcal{I}_{val}^{max} \leftarrow \max(\{\|\mathcal{I}_{val}(\mathbf{x}_i^{tr}, y_i^{tr})\|\}_{i=1}^n)$
**for** $t = 1$ **to** $T$ **do**
    $\mathcal{B} \leftarrow \text{SampleMiniBatch}(D_{train}, r)$ (a mini batch of $r$ data points)
    $\hat{\beta}_\mathbf{x} \leftarrow -\beta \cdot \mathcal{I}_{val}(\mathbf{x}_i^{tr}, y_i^{tr})/\mathcal{I}_{val}^{max}$
    $\mathcal{L} \leftarrow \frac{1}{r} \sum_{\mathbf{z} \in \mathcal{B}} \left( \mathcal{L}_{ce}(\mathbf{x}_i^{tr}, y_i^{tr}, \theta) + \hat{\beta}_\mathbf{x} \cdot E_\theta(\mathbf{x}_i^{tr}) \right)$
    $f_\theta \leftarrow f_\theta - \alpha \cdot \nabla_\theta \mathcal{L}$   (one SGD step)
**end for**
**Return** $f_\theta$

---

As the loss decreasing, $-\frac{\partial E_\theta(\mathbf{x})}{\partial \theta}$ will increase the energy while $-\frac{\partial f_\theta(\mathbf{x})[y])}{\partial \theta}$ will decrease the the energy by increasing $f_\theta(\mathbf{x})[y]$.

$\square$

Remark A.1 tells that optimizing the classifier would lead to an unstable shift of the energy value.

### A.3 PROOF FOR REMARK 3.2

*Proof.* Similar to the proof for Proposition 3.1, for the optimal classifier $f_\theta^*$ and an arbitrary classifier $g_\upsilon$, we could define a classifier $h_\xi$ as:

$$h_\xi(\mathbf{x}) = f_\theta^*(\mathbf{x}) - E_\upsilon(\mathbf{x}) + E_\theta(\mathbf{x}). \tag{20}$$

Then for the predicted probability:

$$\begin{aligned} \bar{p}_\xi(y|\mathbf{x}) &= \frac{\exp\left[f_\theta^*(\mathbf{x})[y]) - E_\upsilon(\mathbf{x}) + E_\theta(\mathbf{x})\right]}{\sum_i \exp\left[f_\theta^*(\mathbf{x})[i]) - E_\upsilon(\mathbf{x}) + E_\theta(\mathbf{x})\right]} \\ &= \frac{\exp\left[f_\theta^*(\mathbf{x})[y])\right]}{\sum_i \exp\left[f_\theta^*(\mathbf{x})[i])\right]} \\ &= \bar{p}_\theta(y|\mathbf{x}). \end{aligned} \tag{21}$$

For the energy $E_\xi(\mathbf{x})$ on the $h_\xi$, we have:

$$\begin{aligned} E_\xi(\mathbf{x}) &= -\log \sum_i \exp\left[h_\xi(\mathbf{x})[i]\right] \\ &= -\log \sum_i \exp\left[f_\theta^*(\mathbf{x})[i] - E_\upsilon(\mathbf{x}) + E_\theta(\mathbf{x})\right] \\ &= -\log\left(\exp[E_\theta(\mathbf{x}) - E_\upsilon(\mathbf{x})] \cdot \sum_i \exp\left[f_\theta^*(\mathbf{x})[i]\right]\right) \\ &= E_\upsilon(\mathbf{x}) - E_\theta(\mathbf{x}) - -\log \sum_i \exp\left[f_\theta^*(\mathbf{x})[i]\right] \\ &= E_\upsilon(\mathbf{x}). \end{aligned} \tag{22}$$

$\square$

## B EXPERIMENT DETAILS

### B.1 DETAILS FOR THE CALCULATION OF INFLUENCE FUNCTION

The calculation of influence function requires a validation set. For imbalanced CIFAR10 and CIFAR100, we sample data points of each class from the class-imbalanced training set to compose the validation set. While the number of data points per class is determined by the minimum number of data points per class in the training set. For ImageNet-LT, since it have a val split, we use the val split to calculate the influence function. For iNaturalist, since the minimum number of data points per class is APPall (2 images per class), we take the *few-shot* classes of the trainig set as the validation set.

We calculate the influence function with stochastic estimation (Cook & Weisberg, 1982) following Koh & Liang (2017). We implement the calculation of influence function based on the Python package for calculating influence function (Lo & Bae, 2022). For imbalanced CIFAR10 and imbalanced CIFAR100, the influence function is calculated with stochastic estimation for 5000 iteration and averaged over 10 trails. For ImageNet-LT and iNatrualist, we calculate the influence function only on the classifier, and theinfluence function is calculated with stochastic estimation for 1000 iteration and averaged over 10 trails.

### B.2 DETAILS FOR THE EXPERIMENTS ON IMBALANCED DATASET

We follow the setting in Cao et al. (2019) to train ResNet-32 on imbalanced CIFAR dataset and report the performance of the model at the final epoch. The model is trained for 200 epochs with SGD optimizer where learning rate at $0.1$, momentum at $0.9$ and weight decay at $2e-4$. The learning rate is decayed with factor $0.01$ at 160-th epoch and 180-th epoch. For IAER, we finetune the model for 5 epochs with batch size at $128$ and learning rate at $1e-4$.

For ImageNet-LT and iNaturalist 2018, we employ the pretrained model provided in Kang et al. (2020) and follow the setting in it to finetune the ResNeXt-50 (Xie et al., 2017) on ImgaeNet-LT and the ResNet-152 (He et al., 2016) on iNaturalist 2018. For ImageNet-LT, the classifier is finetuned for $10$ epochs with batch size at $512$ and learning rate at $0.2$. For iNaturalist, the classifier is finetuned for $30$ epochs with batch size at $512$ and learning rate at $0.2$.

The $\beta$ in Algorithm 1 is searched in $\{0.1, 0.5, 1, 10\}$ for CIFAR-LT and set to be $0.5$ for ImageNet-LT and iNaturalist2018. When the absolute value of energy regularization is bigger than the cross-entropy loss the $\beta$ is set to be $\|\frac{\mathcal{L}_{ce}(\mathbf{x}_i^{tr}, y_i^{tr}, \theta)}{E_\theta(\mathbf{x}_i^{tr}) \cdot \mathcal{I}_{val}(\mathbf{x}_i^{tr}, y_i^{tr})/\mathcal{I}_{val}^{max}}\|$

### B.3 DETAILS FOR THE EXPERIMENTS FOR DOMAIN GENERALIZATION

We follow the setting in Gulrajani & Lopez-Paz (2021) to conduct experiments and use the training domain validation set to calculate the influence function. The $\beta$ in Algorithm 1 is set to be $0.1$. We employ the same training settings and hyper parameters as implemented in Gulrajani & Lopez-Paz (2021).

### B.4 COMPARISON BETWEEN THE INFLUENCE OF THE CROSS-ENTROPY AND THE INFLUENCE OF THE ENERGY.

The influence function of the cross-entropy has been widely used in data valuation (Koh & Liang, 2017) and active learning (Liu et al., 2021). As shown in Fig. 4, the influence of the cross-entropy and the influence of the energy are not correlate with each other. The influence function of the cross-entropy as in previous works (Koh & Liang, 2017; Liu et al., 2021) evaluates the influence of reweighting the data points. As for the influence function of the energy, we focus on the influence of the energy regularization, which acts as reweighting and margin control. Therefore there are fundamental differences between the influence of the cross-entropy and that of the energy.

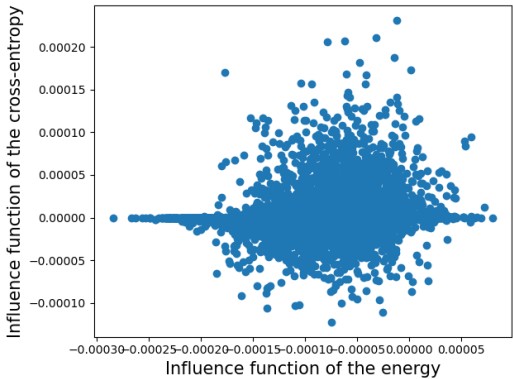

Figure 4: Relation between the influence of the cross-entropy and that of the energy. The influence is calculated for the ResNet-32 trained on CIFAR10 where each point represent a data sample.

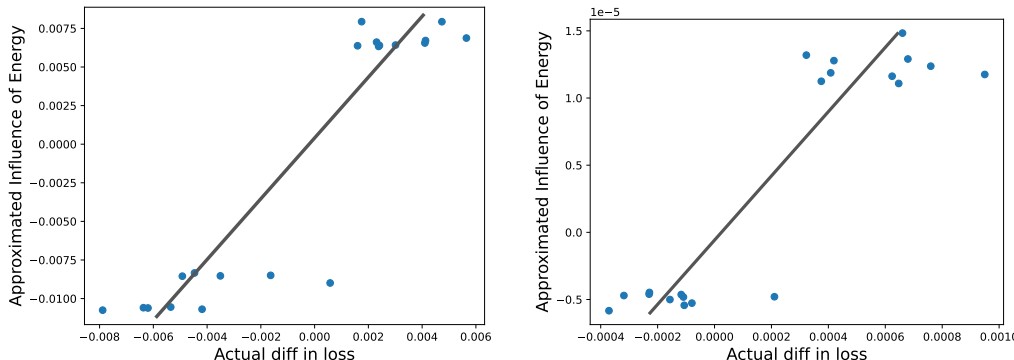

(a) ERM pretrained model (Pearson's R = 0.9154)    (b) LDAM pretrained model (Pearson's R = 0.9037)

Figure 5: The positive relation between the calculated influence function and the actual change in testing loss on pretrained CNN. We plot the top 20 most influential data points.

## C    ADDITIONAL EXPERIMENT RESULTS

### C.1    ADDITIONAL RESULTS TO VALIDATE THE INFLUENCE FUNCTION OF ENERGY REGULARIZATION

In addition to Fig. 5 where we calculate the average influence over the whole testing set, we arbitrary pick a wrongly classified data point and calculate its influence function following Koh & Liang (2017).

We calculate the influence function for the ResNet-32 trained with ERM on the long-tail CIFAR10 and long-tail CIFAR100 where the imbalance ratio is set to be 100. We plot the influence of energy regularization and the actual change in testing loss after finetune the model with energy regularization for 50 epochs on 100 most influential data point. The calculated influence function has a positive relation to the actual change in loss (Pearson's R is 0.7875 on long-tail CIFAR10 and is 0.5744 on long-tail CIFAR100)

### C.2    ENERGY DISTRIBUTION SHIFTS DURING TRAINING

We plot the energy distribution of the training set during the training of ResNet-32 by ERM on the long-tailed CIFAR10 with imbalance ratio at 100. As shown in Fig. 6, the energy distribution keeps changing during the training even thought the training loss is stable *e.g.* from 100-th epoch to 150-th epoch. We further plot the distribution at 160, 170, 180, 190 and 200 epoch after the learning rate have decayed. As shown in Fig. 6(b), the energy distribution of the training set still changes when the learning rate is decayed and the model is converged.

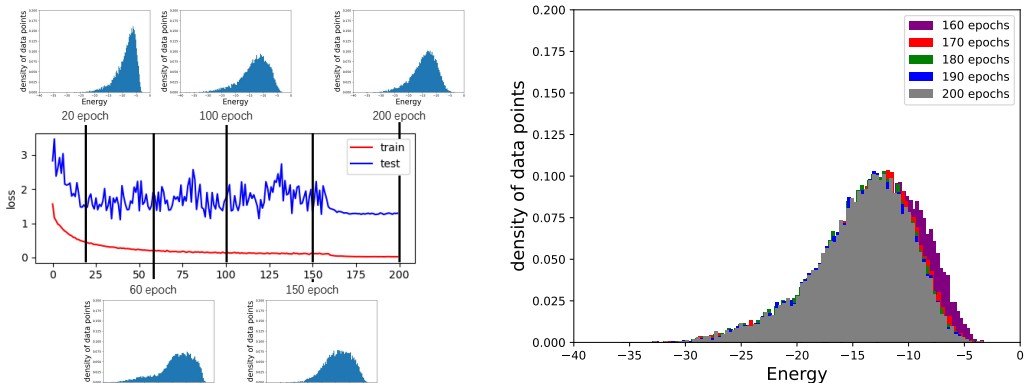

Figure 6: The energy distribution of a ResNet-32 trained on the long-tailed CIFAR10. (a): the energy distribution of the training set on different epochs during the taining (b): the energy distribution of the training set at 160, 170, 180, 190 and 200 epoch.

Table 4: Time required to approximate the influence function on different dataset for different model.

| Model | Dataset | iteration | repeat times | time used |
|---|---|---|---|---|
| ResNet-32 | CIFAR10 | 5000 | 10 | 718.20s |
| ResNeXt-50 | ImageNet-LT | 1000 | 10 | 8492.92s |
| ResNet-152 | iNaturalist 2018 | 1000 | 10 | 9701.74s |

Table 5: The detailed results for domain generalization where we report the testing accuracy(%) for different test domains.

| Method | Testing accuracy on different test domains(%) | | | |
|---|---|---|---|---|
| Colored MNIST | +80% | +90% | -90% | |
| ERM | $72.3 \pm 0.0$ | $72.0 \pm 0.1$ | $10.1 \pm 0.1$ | |
| ERM +IAER | $73.8 \pm 0.1$ | $71.5 \pm 0.1$ | $10.6 \pm 0.1$ | |
| PACS | A | C | P | S |
| ERM | $85.6 \pm 0.1$ | $79.7 \pm 0.2$ | $98.9 \pm 0.1$ | $78.0 \pm 0.1$ |
| ERM + IAER | $84.2 \pm 0.1$ | $84.8 \pm 0.1$ | $97.3 \pm 0.1$ | $80.2 \pm 0.1$ |

## C.3 TIME COMPLEXITY ANALYSIS

The main overhead of the proposed IAER is calculating the influence function of energy regularization. Since we approximate the influence function using stochastic approximation, the time used to calculate the influence function is determined by the choice of hyper parameters. We test the time cost for calculating the influence function for ResNet-32 on CIFAR10 with Intel(R) Xeon(R) CPU E5-2678 v3 @ 2.50GHz and one GeForce RTX 2080Ti for 5000 iteration and averaged for ten times. Approximating the influence function for 5000 iteration takes 718.20s on average. As shown in Table 4, we report the time used to approximate the influence function on different datasets for different models with the same hardware settings.

## C.4 DETAILED RESULTS FOR DOMAIN GENERALIZATION

We report the detailed results of our experiments for domain generalization, as shown in Table 5.

