# OpenReview forum: "Addressing Covariate Shifts with Influence Aware Energy Regularization"
_ICLR.cc/2024/Conference — ICLR 2024 Conference Withdrawn Submission_

### Official Review · Reviewer_YSaa · 2023-10-20

**Soundness:** 2 fair
**Presentation:** 2 fair
**Contribution:** 2 fair
**Rating:** 3
**Confidence:** 2

**Summary:**

The paper proposes techniques for covariate shift adaptation based on energy regularization. In addition, the paper presents multiple experimental results backing the methods proposed.

**Strengths:**

The research topic addressed is very relevant since distribution shifts between training and testing stages are very common in practice and can significantly affect performance of learning methods. In addition, the multiple experimental results carried out can help understand the relevance of the energy function

**Weaknesses:**

The methodological contribution in the paper is unclear. Firstly, the paper seems to focus on covariate shift adaptation but the contribution with respect to a well-developed state-of-the-art for covariate shift adaptation is not described or assessed. In particular, the experimental results do not seem to correspond with covariate shift adaptation. The theoretical results in the paper like proposition 3.1 seem straightforward and not very interesting. In addition, statements such as "by pushing the distribution corresponding to the energy closer to the testing data distribution, the classifier could generalize better to the testing data distribution" are unclear.

**Questions:**

would not be more meaningful for the paper to carry out experimentation in settings affected by covariate shift? and provide comparisons with methods for covariate shift adaptation?

also, describing the contribution with respect to the state-of-the-art for covariate shift adaptation would be useful

---

### Official Review · Reviewer_ibKz · 2023-11-01

**Soundness:** 2 fair
**Presentation:** 2 fair
**Contribution:** 2 fair
**Rating:** 3
**Confidence:** 4

**Summary:**

The paper claims that the energy function of covariate likelihood E(x), when regularized, will help in handling covariate shift. Further, instead of minimizing an average of it over the training set, it is proposed to minimize a weighted average of this. The weights are computed using influence functions computed using a validation set. Simulations on domain generalization and class imbalance benchmarks are performed to show that the proposed methodology improves generalization.

**Strengths:**

1. Covariate shift is indeed an important problem of pragmatic significance.

**Weaknesses:**

1. It is claimed that the intuition for the methodology is that energy function is related to covariate marginal p(x), which plays an important role in covariate shift. While this is correct at a high level, i) the regularizer used is weighted average of p(x^tr). ii) the weights are based on influences over validation loss. However, the validation set is a sample from the source domain. Hence the regularizer essentially maximizes the log-(unnormalized)-likelihood of p(x^tr) (source domain). So it is not clear why this regularizer may help covariate shift ?

2. Results of pro3.1 and  Lemma3.2 seem quite straightforward, essentially trying to prove that conditionals could be same while marginals may be different. I am not sure if they add any insight to the present work.

3. I agree that Eq12 implies there is sample-reweighting happening. However, while classical methods employ the covariates from target domain (unlabelled), the proposed uses influence based weights on validation set, which is from the source domain. So it is not clear why this may help in covariate shift.

4. Results in section4.1 seem a bit out of place because the test/validation is from the source domain. So we already know that influence functions reflect the correlation. I do not think a separate evaluation is required for this. If the same is shown for test/validation loss over the target domain, then it may be interesting.

5. The improvements in table3 seem marginal.

6.

**Questions:**

1. The margin interpretation in (12) seems a bit unclear to me. Can this be explained in more detail?
2. Also, assuming the "margin" in (12) is the coefficient (p(y/x)- 1/(1-\beta(x))), irrespective of \beta >0 or <0 , it seems to decrease. So I am missing something here.
3. The influences in fig 2 seem to be negative for both the frequent and rare classes. So why is there supposed to be a pullup effect on one and pull down on the other ?

---

### Official Review · Reviewer_NQup · 2023-11-04

**Soundness:** 2 fair
**Presentation:** 3 good
**Contribution:** 2 fair
**Rating:** 3
**Confidence:** 2

**Summary:**

The paper proposes to use an energy function of the covariates to improve robustness to covariate shift. The main idea is to define the influence of a function of the covariates which is hypothesized to the be a cause for performance drops under covariate shift. The influence  is then turned into a regularized which is used to fine tune a classifier using Influence-Aware-Energy-Regularizaiton (IAER). Experiments suggest method could improve over baselines on some datasets.

**Strengths:**

1. The experiments seem encouraging in that they improve over existing methods.
2. The proposed algorithm is not too complicated given a subroutine for influence-computation, and it seems to be applicable to a wide range of methods.
3. The writing was not hard to go through except in one or two cases (algorithm 1 should be in the main paper and please comment on the trade-offs of using large validation sets in terms of computation and performance).

**Weaknesses:**

My main concern with this paper is that the proposed method is supposed to target covariate shift (see page 3) but the experiments largely contain long-tail classification, which is a label imbalance problem. Formally, covariate shift and label shift are different kinds of shifts that rely on different kinds of corrections. Maybe I am missing something but the authors need to make it clear how their method really solves the label imbalance problem when their weights and margin corrections do not depend on Y.

Further, CMNIST, VLCS and PACS cannot be purely classified into label or covariate shift. They is style shift in PACS which can take the test distributions completely out of the support of the training, and CMNIST has shifting spurious correlations. How is the proposed method supposed to handle those, especially when run with ERM? Including these experiments gave me a lot of pause beyond just the conceptual issues because it is unclear whether IEAR is actually better.

The other image experiments seem clean but I have one question. Why not do the Imagenet/iNaturalist experiment with all the class sets combined? What is the virtue of doing the evaluations separately? It seems that each validation set is improving only the correspond test set but worse than cRT/LWS. This seems to point to the fact that the proposed method requires more care to run than the baselines.

**Questions:**

Two lemmas are proved to show that classification performance should not be affected by changing the defined energy. But then the authors propose a regularization. Is the takeaway here that the problems that occur in the dataset considered purely due to optimization issues? If that is the case, can the authors connect the method to others like LDAM and importance weighting either formally or intuitively? What kind of differences exist in how different samples are weighted between these methods?


Also, see weaknesses.